# Global assessment of rural-urban interface in Portugal related to land cover changes

Marj Tonini[1], Joana Parente[2], Mario Pereira[2,3]

[1]Institute of Earth Surface Dynamics (IDYST), Faculty of Geosciences and Environment, University of Lausanne, Lausanne, 1000, Switzerland

[2]Centro de Investigação e de Tecnologias Agro-Ambientais e Biológicas (CITAB), Universidade de Trás-os-Montes e Alto Douro (UTAD), Vila Real, 5000-801, Portugal

[3]Instituto Dom Luiz (IDL), Faculdade de Ciências da Universidade de Lisboa, Portugal

*Correspondence to: Marj Tonini (marj.tonini@unil.ch)*

**Abstract.** The rural-urban interface (RUI), known as the area were structures and other human development meet or intermingle with wildland and rural area, is at present a central focus of wildfire policy and its mapping is crucial for wildfire management. In the Mediterranean basin, humans cause the vast majority of fires and fire risk is particularly high in the proximity of infrastructures and of rural/wildland areas. RUI's extension changes under the pressure of environmental and anthropogenic factors, such as urban growth, fragmentation of rural areas, deforestation and, more in general, land use/land cover changes (LULCC). As other Mediterranean countries, Portugal experienced significant LULCC in the last decades in response to migration, rural abandonment, ageing of population and trends associated to the high socioeconomic development. In the present study, we analysed the LULCC occurred in this country in the 1990 – 2012 period with the main objective of investigating how these changes affected RUI's evolution. Moreover, we performed a qualitative and quantitative characterization of burnt areas within the RUI in relation the observed changes. Obtained results disclose important LULCC and reveal their spatial distribution, which is far from uniform within the territory. A significant increase in artificial surfaces was registered nearby the main metropolitan communities of the northwest, littoral-central and southern regions, whilst the abandonment of agricultural land nearby the inland urban areas led to an increase of uncultivated semi-natural and forest areas. Within agricultural areas, heterogeneous patches suffered the greatest changes and were the main contributors to the increase of urban areas; moreover, this land cover class, together with forests, were highly affected by wildfires in terms of burnt area. Finally, from this analysis and during the investigated period, it appears that RUI increased in Portugal more than two thirds, while the total burnt area decreased one third; nevertheless, burnt area within RUI doubled, which emphasizes the significance of RUI monitoring for land and fire managers.

## 1 Introduction

Mediterranean region is particularly affected by wildfires, mainly as consequence of its type of climate and vegetation cover fire proneness (Pellizzaro et al., 2012; Amraoui et al., 2015). In the European Mediterranean countries, fire incidence has dramatically increased in the last decades and the average total annual burnt area (hereafter, BA) has quadrupled since the 60's (San-Miguel-Ayanz et al., 2012), mainly due to changes in climate and land use (Moreira et al., 2011; Ferreira-Leite et al., 2016). Portugal stands out from this group of countries since it counts the highest number of wildfires and has been the third most affected country in terms of BA in the last three decades (Pereira et al., 2014; San-Miguel-Ayanz et al., 2016). On average, about 95% of wildfires with known causes in Europe during the period 1995 to 2010 (corresponding to about 70% of the total number of recorded events) were associated to human activities (Ganteaume et al., 2013). Only a small percentage of wildfires (e.g. 1% in Portugal and 5% in

Spain) were naturally caused by lightings (Mateus and Fernandes, 2014; Vilar et al., 2016). Wildfires have long been considered a dynamic ecological factor and an efficient agricultural and landscape management tool. In the last decades, the increase of the human population along with the consequent expansion of the urban area and land use/land cover changes (LULCC), made the interfaces between the wildland and the human assets vulnerable to more abundant and extreme wildfires. Therefore, wildfires are

increasingly considered a hazard (Bond and Keeley, 2005; Fernandes and Botelho, 2003; Hardy, 2005; Moreno and Oechel, 2012; Pyke et al., 2010; Van Wagtendonk, 2007), which has motivated governments to implement measures for fires prevention, monitoring and mapping.

In this regard, the investigation of the spatial and temporal patterns of wildfires in a given region is fundamental. Normally, this pattern is not random since it is influenced by the surrounding socio-economic and environmental factors. For example, clusters

of wildfires were highlighted in Portugal, with hot spots concentrated in the north-west and center regions, while the southern region presented lower densities of wildfires (Pereira et al., 2015; Tonini et al., 2017). Urban sprawl also affects the spatial-temporal pattern of these hazardous events, making it difficult to set a boundary between human developments and rural areas. The modern urban landscape in Europe has a typical star-shaped spatial pattern (Antrop, 2000) with wedge of unchanged countryside persisting between lobes of urban development (Antrop, 2004). In this context, populations and activities described

either as "rural" or "urban" are closely linked and their distinction is often arbitrary (Tacoli, 1998). In the Iberian Peninsula, the diffuse urban expansion along radiating access roads connecting the center to commercial/industrial zones and isolated housing, as well as the growth of peri-urban centers in previously rural areas, leaves gaps in the suburban/exurban space (Trigal, 2010). Land use changes are at the origin of landscape patterns and dynamics and have a strong influence on forest fires (Silva et al., 2011). On the one hand, each vegetated land cover type, such as agricultural, natural and semi-natural vegetation cover, has a

specific fire proneness depending on the differences in vegetation structure, moisture content and fuel load composition (Barros and Pereira, 2014; Oliveira et al., 2014; Pereira et al., 2014). Further, fire occurrence affects landscape dynamics by changing the vegetation structure and the soil processes according to the fire adaptation of each ecosystem (Viedma, 2008; Pausas et al., 2009). Population and urban areas significantly increased in Europe during the late XX century, which helps to understand the rapid LULCC (Noronha Vaz et al., 2012). In the European Mediterranean countries, LULCC are mainly caused by the increasing

migration to urban centers, at the cost of the abandonment of rural areas, and by the expansion of costal tourism (Alodos et al., 2004; Tedim et al., 2016). One consequence of this process is the urbanization, defined as the process involving the transformation of rural and natural landscape into urban and industrial areas, caused by the interaction of very different factors and largely influenced by communication, accessibility and mobility needs (Antrop, 2000). The dynamic conversion between rural and urban spaces can become extremely complex (Strubelt and Deutschland, 2001): urbanization generates the centralization of certain area

by changing the land use, population density, economical activities and transportation network. This complex process is at the origin of the abandonment of remote rural areas with poor accessibility, which allows the expansion of wild low vegetation and forest (Antrop, 2004). Specifically, the abandonment of low-intensity agricultural lands and grazing practices caused the increase of forest cover and scrubland vegetation (Poyatos et al., 2003; Millington et al., 2007).

Significant LULCC occurred in Portugal in the recent period. The urbanization of coastal areas in the country occurred in

concomitance with the abandonment of agricultural land in marginal areas and seemed to prevail between 1990 and 2000 while, in the later period (2000 – 2006), predominated the intensification of agriculture in areas where irrigation was available (Diogo and Koomen, 2012; Van Doorn and Bakker, 2007). Pereira et al. (2014) observed that among the southern European countries, Portugal registered the highest rates of land use change in the 2000 – 2006 period, marked by a significant increase in artificial

surfaces and Sclerophyllous vegetation and a decrease in forest area and natural grasslands, because of rural abandonment, urbanization and wildfires.

The interface between the wildland and the urban space, called Wildland-Urban Interface (WUI), has been deeply investigated by researchers and fire managers in the last decades. The United States Department of Agriculture (USDA), defined the WUI as the area "where humans and their development meet or intermix with wildland fuels" (Stewart et al., 2007); in this area, fires can spread readily among vegetation fuels and urban structures. Anthropogenic features, such as the distance to roads and houses, have a negative influence on the probability of forest fire occurrence, while the population density positively affects it (Haight et al., 2004; Radeloff et al., 2005; Stewart et al., 2007; Hammer et al., 2009; Lampin-Maillet et al., 2010; Conedera et al., 2015). This trend made these variables to be commonly considered the most important to elaborate WUI maps. Urbanization, and the consequent abandonment of rural areas, caused the expansion of this interface, increasing the probability of wildfires to affect houses and infrastructures (Theobald and Romme, 2007; Zhang et al., 2008). There are strong evidences that the expansion of the urban and WUI area increased the fire density and related risk (Fox et al., 2015; Gallardo et al., 2016; Lampin-Maillet et al., 2010; Viedma et al., 2015), the cost of houses protection from fire (Pellizzaro et al., 2012b), and have an impact on biodiversity and ecosystems (Radeloff et al., 2005). Researchers developed several geospatial models for defining and mapping the WUI, taking into account all the above-mentioned factors. In Europe, Lampin-Maillet et al., (2010) proposed an approach based on the combination of four types of buildings configuration and three classes of vegetation structure. Following this model, Bouillon et al., (2012) developed WUImap, a software tool for mapping WUI in the Mediterranean region successfully applied in southeastern France, eastern Spain and Sardinia, in Italy. In the Alpine context, geospatial approaches to map the WUI were developed in Switzerland by Conedera et al., (2015) and in France by Fox et al., (2015). Pellizzaro et al., (2012b) characterized and mapped the WUI in Sardinia using temporal steps of about 10 years from 1954 and 2008, and found an increase of the WUI's extension. Most of these researches were performed locally, at house-spatial-scale or for small regions within each country, suggesting the need for a homogeneous methodology applicable at national scale. In this regard, Amato et al., (2018) prosed a new procedure based on Multilayer Perceptron and Fuzzy Set Theory to map of the Rural-Urban-Interface for the entire Portugal; this approach allowed to develop continuous non-categorical maps expressing the possibility of being part of this interface in a future scenario. Other studies focused on other aspects related to the WUI, namely hazard/risk, vulnerability and fire risk management in Spain (Badia et al., 2011; Galiana-Martin et al., 2011; Herrero-Corral et al., 2012) as well as fuel and fire modelling in France (Cohen et al., 2003; Pugnet et al., 2013).

The active rural-urban conversion processes and the associated landscape changes, largely documented in the European context, encouraged to reconsider the WUI concept. In this respect, recent studies defined the Rural-Urban Interface (RUI) as an alternative to the WUI, to highlight the importance of including the rural areas, and identified the RUI as the most fire prone areas in Mediterranean countries (Badia-Perpinyà and Pallares-Barbera, 2006; Catry et al., 2010; Moreira et al., 2009). In the present study, the authors investigated the RUI in Portugal with two main objectives: (i) to analyze changes in land use/land cover occurred in this country in the 1990 – 2012 period; (ii) to assess their impact on RUI's evolution. Moreover, the authors performed a qualitative and quantitative characterization of the burnt areas within the RUI in relation to the LULCC. Finally, this research provides a map of the RUI's extension and evolution in the last twenty years (from 1990 to 2012) for the entire continental Portugal.

## 2 Study area

Continental Portugal (c.a. 90,000 km$^2$) is located in the southwest of the Iberian Peninsula, bathed by the Atlantic Ocean on the south and west coast and bounded by Spain at north and east. According to the census data, the Portuguese population was about 10.6 million in 2011 and decreased to 10.3 million in 2017; its distribution displays a much higher density in the northwestern and southern coastal areas as well as around the major cities (Fig. 1). The Tagus river divides the country into two regions of approximately the same area but very different in terms of several biophysical and human drivers (Fig. 1). The north region has a temperate climate, with dry and warm summer, altitude ranging from sea level to about 2000 m and mean watercourse density of about 0.65 km/km$^2$. According to CORINE Land Cover (hereafter, CLC) inventory 2012 (EEA, 2016), 54% of north's area is covered by forests and scrublands, 40% is used for agriculture and about 5% is occupied by artificial surfaces (Fig. 2). The region south of Tagus river is characterized by temperate climate with dry and hot summer, low altitude range (between sea level and about 1000 m), and mean watercourse density of about 0.58 km/km$^2$. According to CLC 2012 inventory, agricultural areas occupy 56% of this territory, forest and semi-natural areas 40%, and artificial surfaces only 2% (Fig. 1).

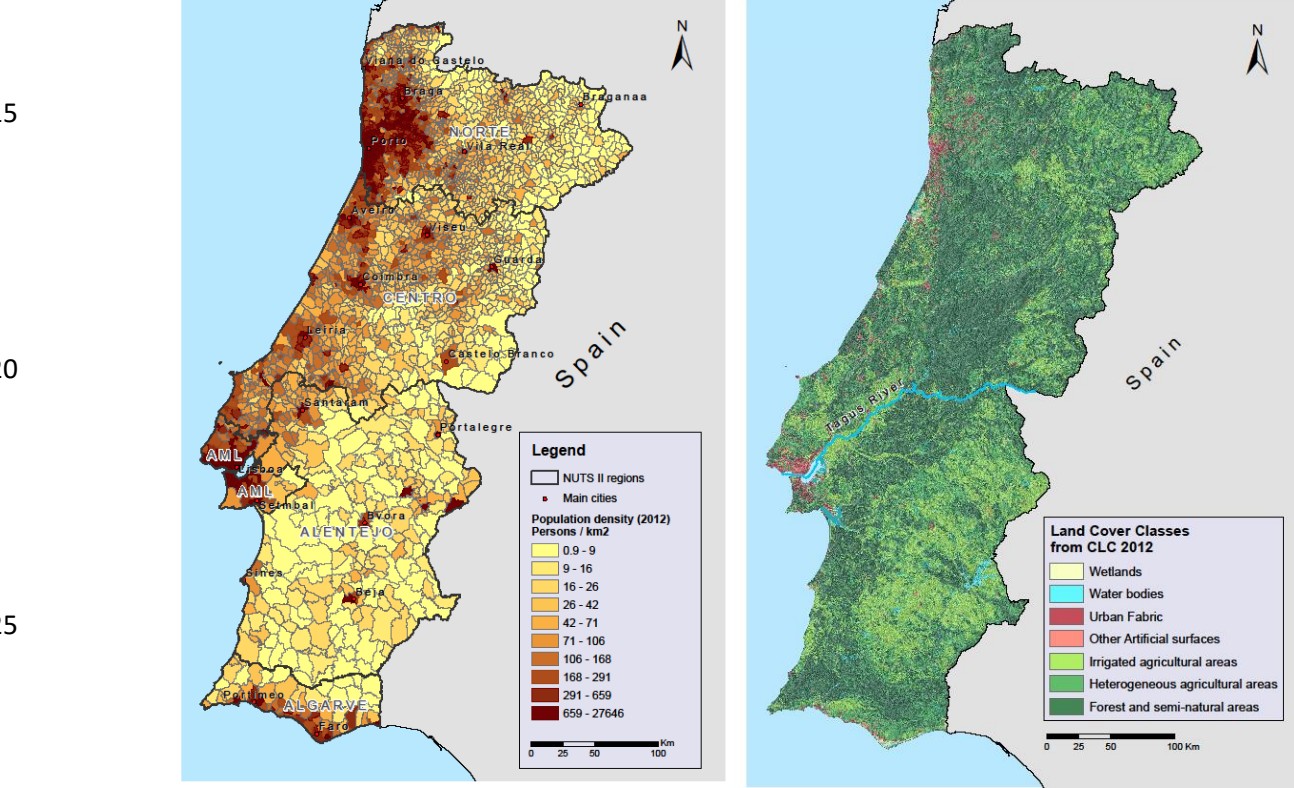

**Figure 1 – (Left) Population density at parish level (INE, 2012) in the mainland Portugal, with the location of the of the major cities. NUTS refers to regions according to Nomenclature of Territorial Units for Statistics level II (Eurostat, 2017; Santos, 2014). (Right) Land cover of mainland Portugal based on the second level of CORINE land cover inventory 2012 (EEA, 2016)**

## 3 Data and methodology

Wildfires digital data for the period 1990 – 2015 came from the National Mapped Burned Area dataset (NMBA), provided by the Portuguese Institute of Conservation of Nature and Forests (ICNF). The Institute of Agronomy (*Instituto Superior de Agronomia*, ISA) produced the official database in the earlier years (1975 to 1989) while the official National fire database covers the

1975 – 2015 period and comprises about 49,000 fire events for a total BA of 4,430,000 ha (Oliveira et al., 2012; Barros and Pereira, 2014). The annual fire perimeter maps resulted from a semi-automatic supervised image classification procedure (Gorte, 1999) performed with the classification and regression trees algorithm of Breiman et al., (1984) on late summer-autumn Thematic Mapper/Enhanced Landsat satellite imagery. Technological improvements of satellite sensors allowed acquiring and processing higher-resolution images with an increasing resolution from the initial 30 hectares to 5 hectares after 1984, and even higher after 2005. To ensure the accuracy of the dataset, results were compared against field statistics gathered on the ground by the National Forest Authority and by the National Civil Protection Authority. For each fire record, the dataset comprises the BA (perimeter map) and the year of occurrence.

Land use/land cover's information came from the CLC inventory provided by European Environment Agency (EEA). CLC is delivered as cartographic product, both in raster (i.e. a regular grid of cells) and vector-shapefile (as point, line and polygons) format. The minimum cartographic unit is of 25 ha (500 by 500 m) with a minimum geometric accuracy of 100 m and a thematic accuracy over 85% (EEA, 1994). CLC nomenclature is a three-level hierarchical classification system with 44 classes at the third and most detailed level (Table 1). The five more general classes for the first level are the following: Artificial Surfaces (AS), Agricultural Areas (AA), Forest and Semi-Natural Areas (FSNA), Wetlands, and Water bodies. CLC inventories are currently available for four periods (1990, 2000, 2006, and 2012) with a minimum time consistency of plus/minus one year. CLC was already used for land-use change and urban dynamics studies in Portugal (Noronha Vaz et al., 2012; Pereira et al., 2014).

In order to identify and detail the major habitats/plant communities/vegetation types corresponding to each CLC class in Portugal, we used the Soil Use and Occupancy Chart (*Carta de Uso e Ocupação do Solo*, COS) provided by the Portuguese Directorate-General of the Territory (*Direção-Geral do Território*, DGT). COS is a national cartographic product with a minimum cartographic unit of 1 ha. We compared CLC2006 with COS2007v2.0 because these are the closest inventories, in time, within the investigated period. In addition, COS2007v2.0 presents improvement from the thematic and geometric point of view (Sarmento et al., 2016): it includes 225 classes (32 more than the initial version) at the most detailed level, distributed over 5 hierarchical levels.

In the present study, the four CLC inventories were employed to analyze LULCC at different levels and to map RUI at different periods, according to the methodology described below. The first analysis consisted in investigating LULCC within the entire the study period (i.e. from 1990 to 2012). This allowed elaborating the map of changes showing the transitions among the five more general classes (CLC first level hierarchy) and, more in detail, to quantify gained and lost areas with reference to both the first and the second level hierarchy of CLC. Moreover, the difference between gains and losses within each class divided by the total area covered by the specific class in the later period, representing the net change, was computed and the result expressed in percentage. RUI was then mapped for each period (1990, 2000, 2006, and 2012) using a geospatial approach designed to extract the area of intersection between a buffer around the AS and the area resulting from the sum of the FSNA plus the Heterogeneous Agricultural Areas (HAA, a sub-level of AA). Different buffer widths from 100 m to 2 km were tested. Finally, we adopted a buffer width of 1 km, corresponding to two times the spatial resolution of CLC inventories. This value is in line with the values used in other countries for WUI mapping (Radeloff et al., 2005; Vilar et al., 2016) and, at the same time, is large enough to avoid bias in the results, due to the CLC spatial resolution. The other agricultural areas (i.e. arable lands, permanents crops and pastures) were not included in the RUI definition since these vegetated land covers are usually well managed, mostly irrigated and frequently constitute an obstruction to fire spread. Similarly, San-Miguel-Ayanz et al., (2012) suggested that only HAA have to be considered in the definition and quantification of the RUI in Portugal, together with FSNA.

The geocomputation allowing the RUI mapping was performed under ArcGIS™ software environment. Namely, the geoprocessing workflow was implemented into a Model Builder (Fig. 2), a specific application used to create, edit and manage models, meant as workflows that string together sequences of geoprocessing tools (e.g. selection, buffer, intersect), feeding the output of one tool into another tool as input (i.e. the raster or vector digital data). Finally, we analyzed how each land cover class (considering the third level hierarchy of CLC) was affected by wildfires in terms of BA for each investigated period within the RUI. To this end, polygons defining the BA registered at each CLC_year plus/minus one (1989 – 1991, 1999 – 2001, 2005 – 2007, 2011 – 2013) were merged together and the resulting BA polygons were clipped over the corresponding RUI map. The resulting outputs, representing the BA within the RUI cumulated over three years around each investigated period, were finally overlapped with the CLC source map.

| Level 1 | Level 2 | Level 3 |
|---|---|---|
| 1 Artificial surfaces | 11 Urban fabric | 111 Continuous urban fabric |
| | | 112 Discontinuous urban fabric |
| | 12 Industrial, commercial and transport units | 121 Industrial or commercial units |
| | | 122 Road and rail networks and associated land |
| | | 123 Port areas |
| | | 124 Airports |
| | 13 Mine, dump and construction sites | 131 Mineral extraction sites |
| | | 132 Dump sites |
| | | 133 Construction sites |
| | 14 Artificial, non-agricultural vegetated areas | 141 Green urban areas |
| | | 142 Sport and leisure facilities |
| 2 Agricultural areas | 21 Arable land | 211 Non-irrigated arable land |
| | | 212 Permanently irrigated land |
| | | 213 Rice fields |
| | 22 Permanent crops | 221 Vineyards |
| | | 222 Fruit trees and berry plantations |
| | | 223 Olive groves |
| | 23 Pastures | 231 Pastures |
| | 24 Heterogeneous agricultural areas | 241 Annual crops associated with permanent crops |
| | | 242 Complex cultivation patterns |
| | | 243 Land principally occupied by agriculture, with significant areas of natural vegetation |
| | | 244 Agro-forestry areas |
| 3 Forest and semi natural areas | 31 Forests | 311 Broad-leaved forest |
| | | 312 Coniferous forest |
| | | 313 Mixed forest |
| | 32 Scrub and/or herbaceous vegetation associations | 321 Natural grasslands |
| | | 322 Moors and heathland |
| | | 323 Sclerophyllous vegetation |
| | | 324 Transitional woodland-shrub |
| | 33 Open spaces with little or no vegetation | 331 Beaches, dunes, sands |
| | | 332 Bare rocks |
| | | 333 Sparsely vegetated areas |
| | | 334 Burnt areas |
| | | 335 Glaciers and perpetual snow |
| 4 Wetlands | 41 Inland wetlands | 411 Inland marshes |
| | | 412 Peat bogs |
| | 42 Maritime wetlands | 421 Salt marshes |
| | | 422 Salines |
| | | 423 Intertidal flats |
| 5 Water bodies | 51 Inland waters | 511 Water courses |
| | | 512 Water bodies |
| | 52 Marine waters | 521 Coastal lagoons |
| | | 522 Estuaries |
| | | 523 Sea and ocean |

**Table 1: CORINE Land Cover (CLC) nomenclature. Numbers on the left represent the CLC code. (Source: EEA, 2016)**

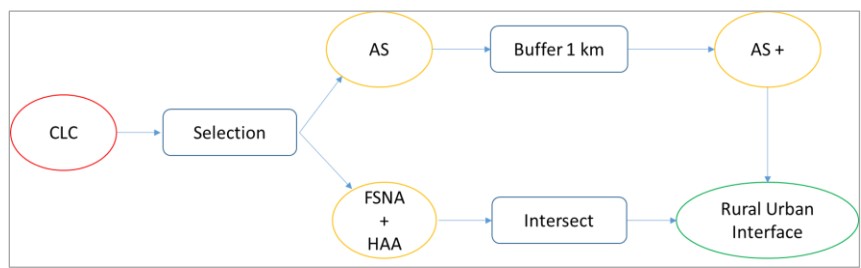

**Figure 2 - Framework implemented in the Model Builder (ArcGIS™) to map the rural-urban interface. CLC=CORINE land cover; AS = Artificial surfaces; FSNA = Forest and Semi natural areas; HAA = Heterogeneous Agricultural areas.**

## 4 Results

### 4.1 LULC change analysis

The analysis of main changes between CLC1990 and CLC2012 in the area occupied by the first level hierarchy classes allowed to visualize the transitions occurred within the entire investigated period (Fig. 3) and thus to have an overview of the LULCC occurred in Portugal. It resulted that major transitions occurred between vegetated areas (i.e. AA and/or FSNA) to AS and between FSNA and AA in both directions. AS increased mainly nearby the main metropolitan communities of the northwest, coastal central and southern regions. A transition from vegetated areas (AA and FSNA) to AS is also visible in center-north and is probably due to the intensification of the main road network to connect the emergent inhabited rural area. The conversion from FSNA to AA and vice-versa appeared to be an active and dynamic process prevailing in the southern half of the country, but it was revealed also in the inner northern region. Figure 4 shows the areas gained and lost for each CLC first-level class and the net percentage of changes, computed relatively to the total area of each class in the later land cover class. The main changes in terms of area were registered by AS, which increased $165 \times 10^3$ ha, and AA, which decreased $184 \times 10^3$ ha, but in terms of net percentage of change the increase of AS was about 50%, while AA decreased only 4.4%. The two classes which manly contributed to the increase in AS were AA, with $110 \times 10^3$ ha, and FSNA, with $50 \times 10^3$ ha.

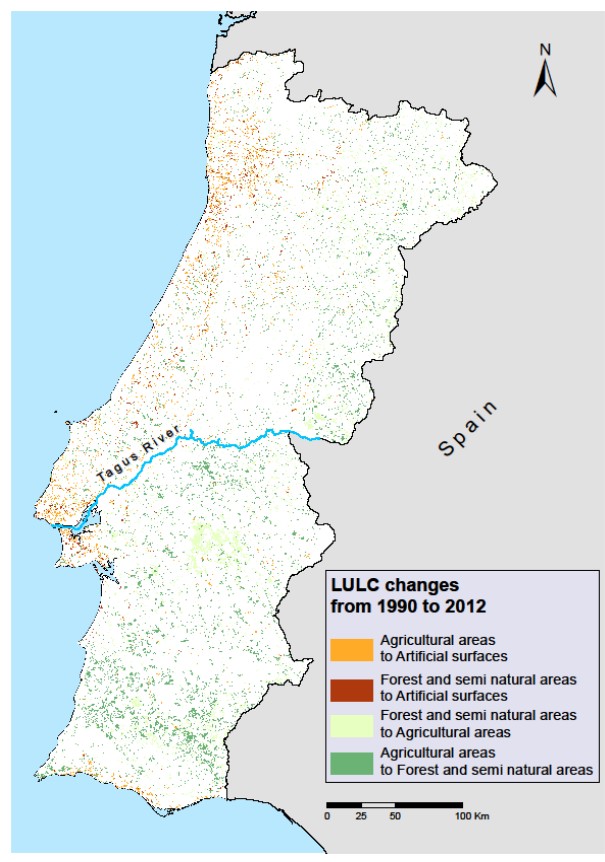

**Figure 3 – Map of land cover/land use transition from 1990 and 2012, evaluated considering the first level hierarchy of CLC 1990 and CLC 2012**

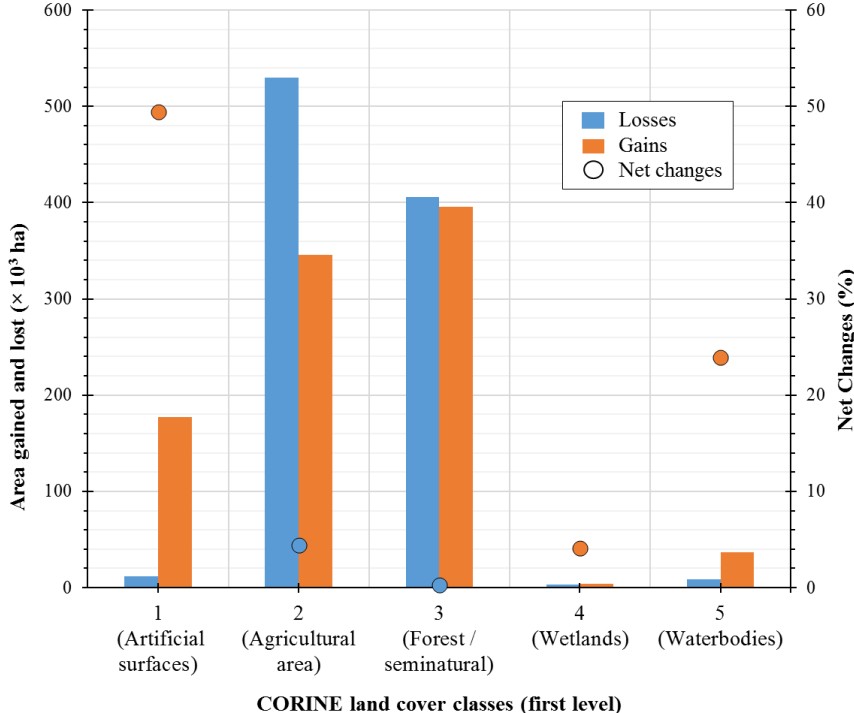

**Figure 4 – Area lost and gained from 1990 to 2012 for each CORINE land cover class, considering the first level hierarchy. Net percentage changes were computed relatively to the total area of each class in the later land cover.**

A more detailed analysis was carried out to investigate changes occurred within classes of the second level of hierarchy from 1990 to 2012. Figure 5 shows that the majority of the CLC classes (level 2, Table 1) displayed important net changes in terms of relative gains and losses compared with values for the same classes in the later period. Scrub and/or herbaceous vegetation associations (code 32) registered a net gain of about 520 ×10³ ha (+24%), while Forest area (code 31) decreased about 460 ×10³ ha (-23%). Arable land (code 21) was the only AA registering an important negative net change of -225 ×10³ ha (-20%). Among AS, Urban fabric (code 11) significantly increased 110 ×10³ ha (45%), and, in terms of net percentage of change by class, all the other three AS sub-levels, including Industrial/commercial and transport unit (code 12), Mine/dump and construction sites (code 13), Artificial/non-agricultural vegetated areas (code 14), increased more than half.

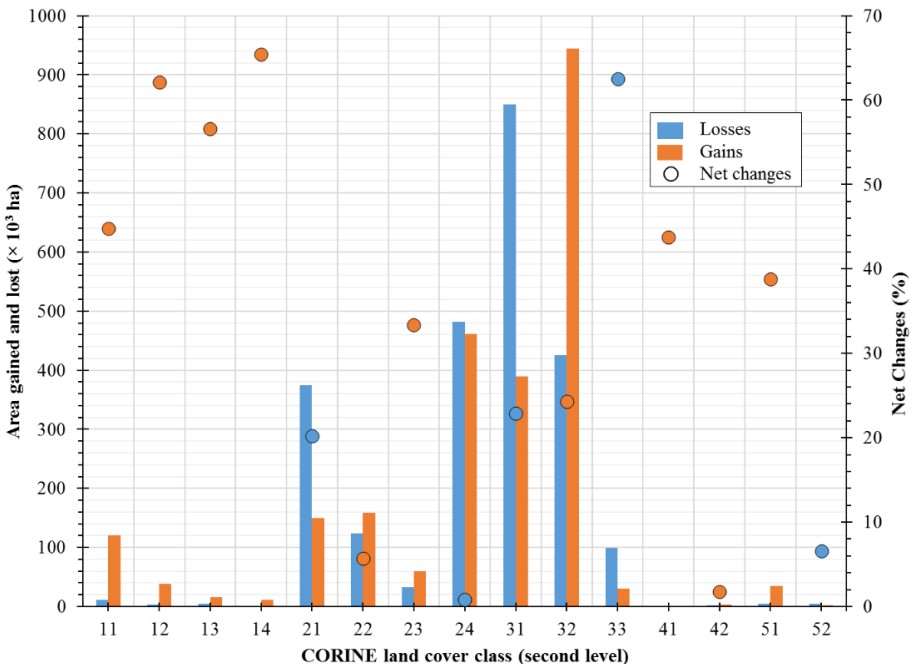

**Figure 5 – Area lost and gained from 1990 to 2012 for each CORINE land cover class, considering the second level hierarchy. Net percentage changes were computed relatively to the total area of each class in the later land cover.**

The bar graph of the contributions to net changes in the AS sub-levels (Fig. 6) shows that Urban fabric (orange bars), which includes buildings, roads and artificially surfaced areas, grew mainly at the expense of HAA (code 24). On the other hand, the increase of Industrial commercial and transport (blues bars) was mainly due to the decrease of Forests (code 31), HAA (code 24) and Scrub and/or herbaceous vegetation associations (code 32).

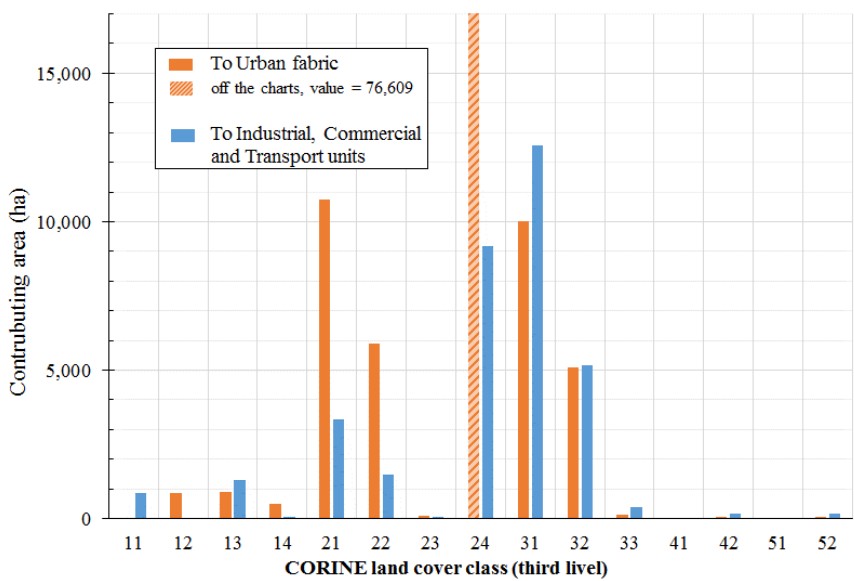

**Figure 6 – Contribution to the net changes from 1990 to 2012 of "Urban fabric" (orange bars) and "Industrial, commercial and transport" (blue bars) from the other CLC sub-levels**

## 4.2 Spatial distribution and characterization of burnt areas

Almost all the CLC third-level classes belonging to FSNA (code 3) were affected by wildfires, in terms of BA (Table 2), with the Transitional woodland-shrub (code 324) and Mixed forest (code 313) as the first and second more damaged classes. This trend was similar in all the four investigated frame-periods, as highlighted in Figure 7, where the same results are expressed in percentage, as the ratio of BA over the total BA for the entire frame period, for each CLC classes considering only the areas within the RUI. The peak of BA in Transitional woodland-shrub (code 324) equals to about $43 \times 10^3$ ha in the 2005 – 2007 period, compared with about $15 \times 10^3$ ha in 2011 – 2013, $14 \times 10^3$ ha in 1999 – 2001 and $6 \times 10^3$ ha in 1989 – 1991. It also emerges that three over four sub-levels of HAA (codes 243, 241, 242) are highly affected by wildfires, thus confirming the need of including HAA in the RUI's definition.

| CLC code | CLC classes | BA within RUI (ha) | | | |
|---|---|---|---|---|---|
| | | 1989 – 1991 | 1999 – 2001 | 2005 – 2007 | 2011 – 2013 |
| 324 | Transitional woodland-shrub | 6086.31 | 8608.30 | 43274.65 | 14638.53 |
| 313 | Mixed forest | 4368.85 | 3282.67 | 4059.12 | 6763.27 |
| 312 | Coniferous forest | 3104.72 | 2055.41 | 1501.67 | 3572.43 |
| 322 | Moors and heathland | 1835.03 | 2329.69 | 3508.50 | 3492.56 |
| 243[*] | Land principally occupied by agriculture, with significant areas of natural vegetation | 1535.36 | 1407.71 | 3252.81 | 3320.70 |
| 311 | Broad-leaved forest | 1144.72 | 1081.55 | 1299.13 | 4234.28 |

| 241* | Annual crops associated with permanent crops | 698.75 | 253.85 | 996.29 | 930.20 |
|---|---|---|---|---|---|
| 242* | Complex cultivation patterns | 677.27 | 688.53 | 1682.33 | 1123.00 |
| 321 | Natural grasslands | 638.39 | 1019.65 | 1180.63 | 1072.76 |
| 323 | Sclerophyllous vegetation | 562.68 | 477.15 | 1386.21 | 314.90 |
| 334 | Burnt areas | 396.87 | 640.29 | 4251.27 | 1691.23 |
| 333 | Sparsely vegetated areas | 321.85 | 761.01 | 1557.09 | 1403.56 |
| 332 | Bare rocks | 123.84 | 157.20 | 99.47 | 11.32 |
| 244* | Agro-forestry areas | 17.25 | 32.19 | 93.43 | 185.92 |
| 331 | Beaches, dunes, sands | 5.22 | 2.65 | 2.65 | 0.99 |

\* Heterogeneous agricultural areas

**Table 2 – Classes of land use, as defined by the third level hierarchy of the CORINE Land Cover (CLC), affected by wildfires expressed in terms of Burnt Area (BA) within the Rural Urban Interface (RUI) during three investigated frame periods.**

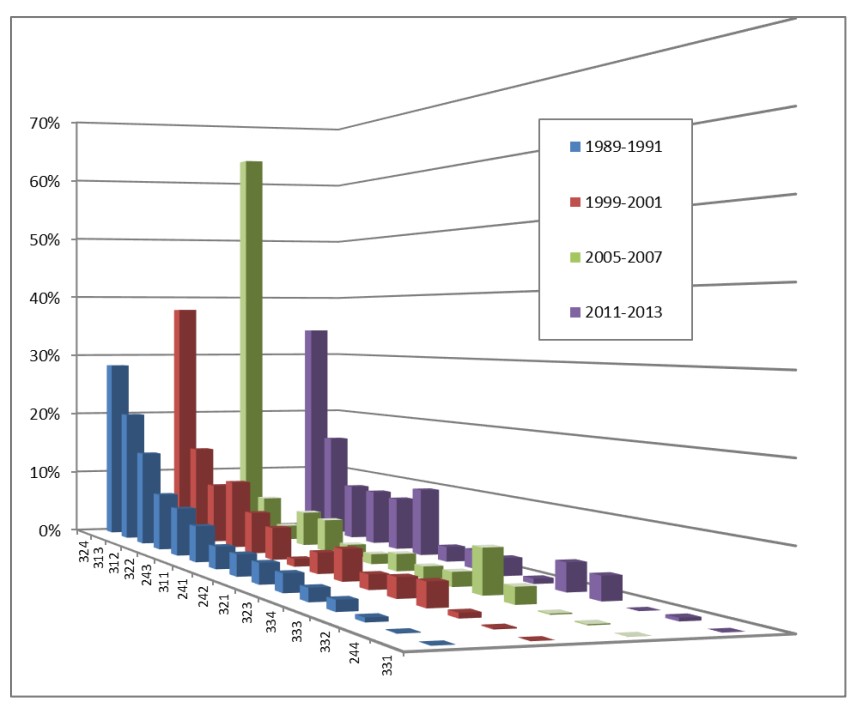

5   **Figure 7 – Classes of land use, as defined by the third level hierarchy of CORINE Land Cover, affected by wildfires expressed in percentage as the ratio of burnt area (BA) affecting each CLC classes over the total BA for each frame period.**

## 4.3 RUI map

In the present study, the classes FSNA and HAA were considered to describe the flammable rural area which intermingling with
10   the urban area defines the RUI. The result was an interface zone evolving in space and in time due to LULCC (Fig. 8). Analyzing

the period 1990 – 2012, the increase of RUI was more active in the north-west and along the coast, where the transition from HAA to Urban fabric was particularly intense. This evolution was mainly due to the urban growth and to the intensification of the road network.

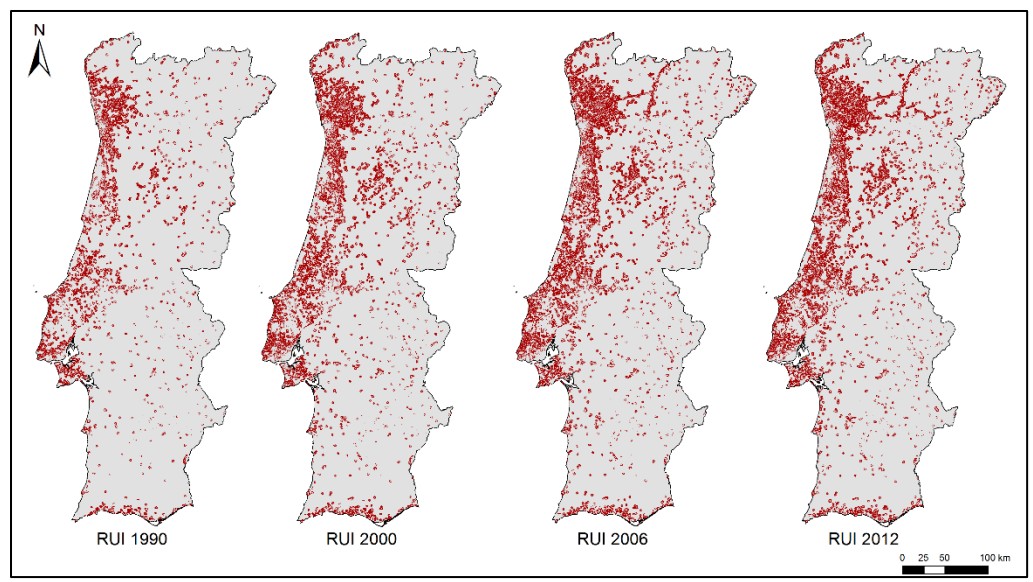

**Figure 8 – Maps of the rural-urban interface (RUI) in Portugal estimated for the different periods of investigation, and corresponding to the available CORINE Land Cover inventories (1990, 2000, 2006, and 2012)**

The total size of the RUI, the fraction of BA within the RUI (RBA) and the total BA (TBA) were evaluated (Fig. 9). It resulted that RUI increased from about $780{\times}10^3$ ha in 1990 up to about $1310{\times}10^3$ ha in 2012 following a power-law (RUI=776310.year^0.1686, $R^2$=0.99). Moreover, we computed the contribution both to the RUI and to the BA within the RUI (BAR) of each CLC class that make up the RUI, namely HAA (code 24), Forests (code 31), Scrub and/or herbaceous vegetation associations (code 32), and Open spaces with little or no vegetation (code 33). Results can be summarized as follows (Fig. 9): (a) the relative contribution of those four CLC classes to the RUI increases in each period almost equally; (b) HAA (code 24) is the CLC class with the largest area within the RUI (~50%); (c) in terms of BAR, the most affected class is the Scrub and/or herbaceous vegetation associations (code 32), followed by Forest (code 31), HAA (code 24) and then Open spaces with little or no vegetation (code 33). TBA values fluctuated, with a maximum in 1990 (about $500{\times}10^3$ ha) followed by 2006 (~ $460{\times}10^3$ ha), while in 2000 and 2012 its value was lower and equal to about $310{\times}10^3$ ha. The RBA, expressed as percentage over the total BA, tends to increase in time, passing from 4% and 7% in 1990 and 2000 to 15% and 14% in 2006 and 2012, respectively.

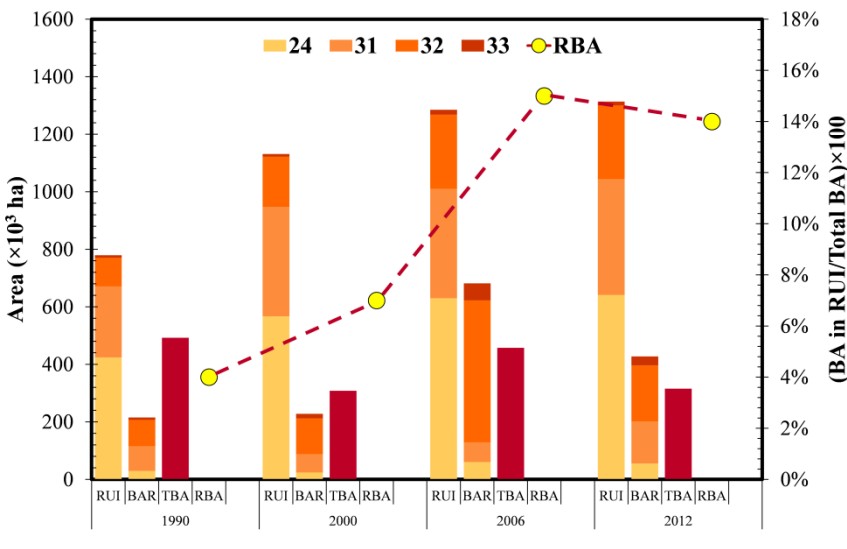

**Figure 9 – Evolution of the absolute value of the area of the rural-urban interface (RUI), burnt area within the RUI (RBA), total burnt area (TBA), and relative burnt area within the RUI (RBA, as %), for each investigated period (1990, 2000, 2006, and 2012).**

**5 Discussion**

The LULCC analysis performed in the present study indicates that from 1990 to 2012 AS (code 1) globally increased in Portugal about 50% (Fig. 4). This growing process is in good agreement with previous findings of other authors (Marques et al., 2014; Meneses et al., 2014; Oliveira et al., 2017; Tavares et al., 2012). Moreover, the present study confirms that the urban growth process in Portugal (quantified as changes in AS) was firstly caused by the transition from HAA (code 24) and secondly from

FSNA (code 3) (Fig. 3 and Fig. 6). The urban development mainly affected the south coastal regions, especially the area between *Portimão* and *Faro*, and was particularly strong nearby to the main metropolitan communities of the northwest and littoral center, namely Porto and Lisbon (Fig. 3 and Fig. 1). Silva and Clarke (2002) described the characteristics and the recent intense urban growth of the metropolitan area of Porto (MAP) and Lisbon (MAL) associated with the economic growth in the end of the XX century (Fernandez-Villaverde et al., 2013). More in details, a mixture of urban surfaces and large farmlands, intense urbanization

along with train lines and main roads, and the emergence of tertiary centers characterized MAL urban pattern. On the other hand, MAP is described by scattered urbanization and dispersed settlements, towns and rural villages surrounded by mountains, within small patches of intensive agriculture and pine forests in a steep slope topography. The decline of the rental market at country level lead to the degradation of old urban areas and to the increase of new constructions in the immediate periphery of Lisbon, while in the north new houses were built by the owners in their small plots of land, promoting a more dispersed urban pattern and an

irregular spatial growth (Silva and Clarke, 2002). The dispersion of the population and of its activities in MAP can also be explained as a consequence of the regional territory planning and the polycentric models of urban growth adopted by the national authorities (Cardoso, 1996; Silva and Clarke, 2002). Another active process identified by the performed change analysis is the abandonment of agricultural lands nearby the inland urban areas, which leads to an increase of uncultivated semi-natural and forest areas (Fig. 3) causing an increase of the urban/rural interface.

As regards the RUI definition and mapping model developed in the present study, different buffer width from 100 m to 2 km were tested, resulting in areas of different size. Vilar et al., (2016) applied a buffer width of 100 m, corresponding to the median of the distances defined in each country's national legislation (Portugal, Spain, South-France and Italy) for protection against wildfires, which makes brush clearing obligatory within a certain radius around each house located close to forests or scrublands. In US, WUI was defined as developed areas in the vicinity of wildland vegetation and mapped considering census blocks above 6.17 housing units/km$^2$ that are within a distance of 2.4 km from wildland vegetation (Radeloff et al., 2005). Finally, we decided to show results obtained applying a buffer width of 1 km because smaller values, even if more in line with the Portuguese national indications, could bias the results, given that the spatial resolution of the CLC inventory was of 500 by 500 meters. RUI definition aimed to map the developed area located in close proximity of wild vegetation, where wildfires can cause deaths, injured, damages to human structures, and finally where human-caused wildfires are more likely to occur. Nevertheless, RUI map was not based on fire incidence measures, thus it not aimed to assess fire risk or fire regimes, which depend on other factors such as topography, climate, weather, vegetation characteristics (Radeloff et al., 2005; Parente and Pereira, 2016). Most of RUI's area detected in Portugal (Fig. 8) was located in regions of high population density and surrounding major cities, while RUI's growths mainly occurred in the transition zones from vegetated lands (AA and FSNA) to AS (Fig. 3). The urbanization process and the consequent reconfiguration of Portuguese cities caused new urban problems and challenges associated with the increased fragmentation of the cities and complex rural-urban relationships, reported for Portugal and Spain (Trigal, 2010). It is important to underline that the impressive increase of the RUI and of BA within the RUI, detected in just a little more than two decades, is not exclusive of Portugal. In Continental US, WUI increased 52% from 1970 to 2000 and 90% of this area included high and highly variable severity fire regimes (Theobald and Romme, 2007). In Europe, Fox et al., (2015) found a progressive increase in fire risk in French Maritime Alps in the period 1960 – 2009. Badia et al., (2011) noticed that two representative Mediterranean WUI areas in Catalonia were more prone to wildfires in the most recent decade of 2000 than in the 1990s. Pellizzaro et al., (2012) analyzed WUI's dynamics and landscape changes in a tourist area of North-East Sardinia (Italy) from 1954 to 2008 and discovered that LULCC was largely associated to a transition from an agro-pastoral economy to one based on tourism. Moreover, they found an increase in the number of houses and dwellers, which tripled during the study period, a sharp grow in summer population, the increase in the length of road network and, finally, the increase of the WUI's extension.

The inspection of the accurate Soil Use and Occupancy Chart national map (COS2007 v2.0) allowed to identify the vegetation types and major habitats/plant communities corresponding to each one of the CLC classes for Portugal. Table 3 of the annex material shows the composition of the CLC classes in terms of COS classes. For simplicity, results are only presented for COS classes with more than 1% of total CLC area. Nevertheless, it is important to underline that these 19 COS classes account for 98.3% of total CLC area. Despite some expected discrepancy, essentially noted in CLC classes less affected by wildfires, results obtained for all the other classes can be summarized as follows: (i) Temporary dryland crops is the larger COS class (22% of total CLC area) and accounts for higher area fraction in CLC Agricultural area (code 2) and Scrub and/or herbaceous vegetation associations (code 32); this is particularly consistent with the agricultural practices, especially in southern Portugal; (ii) COS class of Temporary irrigated crops is well distributed over the CLC classes of Rice fields, Permanently irrigated land, Pastures, as well as Beaches, dunes, sands; (iii) COS Shrub classes are particularly present in CLC classes of Shrubs, Pastures as well as in Open spaces with little or no vegetation; (iv) Tree vegetation in COS is also well related to the correspondent CLC classes and allows to better understand the composition of Forests and Agricultural areas: for example, pure or mixt forests of Cork and Holm oak trees are particularly evident in the CLC classes of Agro-forestry areas, Broad-leaved forests and, Non-irrigated arable lands; on the

other hand, Eucalyptus (pure or mixed forests) in COS are important components of the CLC classes of Mixed forests, Complex cultivation patterns, Annual crops associated with permanent crops and, Broad-leaved forests; finally, pure *pinus pinaster* forests in COS are comprised in the CLC classes of Coniferous forests, Transitional woodland-shrub, and are especially important in Beaches, dunes, sands, where account for 30% of total area; this finding is in good agreement with the presence of *pinus* forests in
the entire central western coast.

It is important to underline that, from 1990 to 2012, the increase in the BA within the RUI is higher (100%) than the increase in the RUI area (70%). This result suggests that other factors, besides the increase of the RUI area, are responsible of the increase of the burnable area within the RUI. In this regard, it is important to take into account some of the specific characteristics of the country, well described in terms of demographic, territory and forest statistics compiled in Feliciano et al., (2015), which can help
to understand the most important factors affecting the forest management in Portugal. Forest is nowadays the dominant land cover in the country (with more than 35% of total area), followed by bushes and grasslands (>29%), and agricultural areas (>24%) (FAO, 2018; Feliciano et al., 2015). According to the National Forest Inventory (IFN, 2010), in the 1995 – 2010 period, four tree species occupied about 85% of total forest area: Eucalyptus (22% – 27%), Cork oak (23% – 24%), Maritime pine (30% – 23%) and, Holm oak (10% – 11%). The first three species have the ability to generate land and business income exceeding 50 €/ha/year (CM, 2015).
Most of the forests and wooded lands (>93% in 1995) have non-industrial private owners and there is a high fragmentation of the forest property, particularly evident in the private sector (Mendes, 2004). Management practices are also very different and changed significantly in the last years, especially in non-industrial private forest (Canadas and Novais, 2014). According to Feliciano et al., (2015), 1/3 of Eucalyptus area is well managed by the industrial pulp and paper companies, with their own forest management and wildfire prevention/fighting resources, while the remaining area is managed by non-industrial private owners, characterized by
different objectives and economic logics. In addition, there is a significant heterogeneity in the spatial distribution of all these characteristics (Baptista and Santos, 2005). Small forest holdings (<10 ha), mainly composed by pine and eucalyptus with low profitability, are much more frequent in the northern and central Portugal, while large properties (>100 ha), essentially of cork oak or a complex and unique agroforestry system of cork oak savanna ("*montado*"), are predominant in the southern regions of the country. Table 4 of the annex material provides a general description of the main characteristics of forest holdings and forest
owners and summarizes the interrelationship between these factors.

Other aspects related to LULCC, such as climate change and biodiversity, are somewhat outside the scope of the present study. However, the abandonment of rural and forest areas and traditional agricultural practices, and the lack of forestry management practices lead to an increase of biomass and fire risk, which can be empowered in a warmer and drier future climate and may have profound impact on ecosystems and biodiversity. For example, the *montado*, which is composed by sparse cork oak trees and a
diversity of understory vegetation (e.g., shrub formations, grasslands), supports higher levels of biodiversity. The decrease in the demand of cork has led to a reduction in management practices and to the abandonment of these lends, leading to the invasion of shrubs, reducing the biodiversity and degrading the services provided by these ecosystems (Bugalho et al., 2011).

Results from our analyses shows that in the second half of the investigated period (2000 – 2006 and 2006 – 2012) the growth rate of RUI was lower than in the first decade (1990 – 2000), probably due to a lower growth rate in the process of urbanization of rural
areas. Moreover, the decrease of relative BA within the RUI from 2006 to 2012 could be associated with the relative decrease of BA in the last investigated period, as a consequence of recent plans for territorial spatial planning and protection of forest against forest fires (Mateus and Fernandes, 2014; Parente et al., 2016). At European level, urban sprawl's complexity and magnitude motivated the European Commission to recommend actions and to coordinate land use policies, within the European Cohesion

Policy 2007-2013 period (CEC, 2006; EEA, 2006). In Portugal, these efforts were complemented with national programs and regional plans such as the National Policy and Territorial Management (*Programa Nacional da Política de Ordenamento do Território*, PNPOT) and the Regional Plan for Territorial Planning (*Plano Regional de Ordenamento de Território*, PROT), supporting the sustainable development and the environmental landscape quality of NUTS-III areas (Noronha Vaz et al., 2012).

However, as far as we know, there is no a specific or general Portuguese legislation about WUI or RUI. It only exists one general mention about RUI in the National Plan to Protect the Forests against Wildfires (CM, 2009). As a measure to protect the urban-forest interface,  this Plan promote the need of creating and maintaing an external buffer strips (10 – 100 m) around population clusters, especially in those with the highest fire vulnerability, as well as around parks, industrial polygons, landfills, housing, shipyards, warehouses, and other buildings.

Finally, we firmly believe that the results of the present study are sufficiently motivating to promote the development of specific policies and legislation, as well as changes in forest and fire management. The detected growth of the RUI area in Portugal from 1990 to 2012, and particularly of the BA within the RUI, clearly suggests the need of improving fire prevention measures and preparedness policies for this interface region. In fact, as indicated by the Portuguese National Fire Plan 2006 (Oliveira, 2005), the increase of the rural urban interface, as a consequence of the above-mentioned LULCC, causes this area to be under the use of

people not educated for fire and unware of possible source of ignition. In particular, Portuguese forestry/forest managers must prioritize sustainable forest management practices and make brush clearing obligatory. These paradigm shifts make even more sense if one takes into account that the fire risk is likely to increase in a future climate associated to a higher frequency of longer and more intense extreme events, such as drought and heat waves. Indeed, results of several recent studies suggest a significant increase of the burnt area, not only in Portugal (Pereira et al., 2013) but in the entire Iberian Peninsula (Sousa et al., 2015), for

different scenarios of future climate and LULCC (Amato et al., 2018) . These findings are in line with the fact that Portugal is the only European country were burnt area did not decrease in the last decades (Turco et al., 2016) and with the increase of the future fire danger for the Mediterranean basin countries of Europe and north Africa (Bedia et al., 2014). In addition, burnt area in Portugal and in the Mediterranean Basin is clearly associated to extreme weather and climate variability, namely the occurrence of heat waves and drought (Amraoui et al., 2015; Pereira et al., 2005; Telesca and Pereira, 2010; Trigo et al., 2006, 2016). Russo et al.,

(2017) showed that drought promotes a synchronous influence on burned areas over a great part of the Iberian Peninsula . Parente et al., (2018) found that all the extreme wildfires in Portugal occurred during or immediately after a heat wave; using projections of an ensemble of climate models, their findings also suggest the increase in the number, duration and amplitude of the heat waves in Portugal for future climate scenarios. An increase of drought severity in the whole Mediterranean basin is expected in the future (Hertig and Tramblay , 2017), suggesting an increase of the fire risk and burned areas, which will impact the LULCC.


**6 Conclusions**

Continental Portugal registered important land cover changes (about 9% of the whole area) and an increase of the rural-urban interface in the investigated period (1990 – 2012). Most significant changes were associated to transitions from the following CORINE Land Cover classes: Agricultural areas (35.1%) and Forest and semi-natural areas (15.2%) to Artificial surfaces

(including Urban areas) and; Agricultural areas to Forest and semi-natural areas (7.3%) and vice-versa (6.3%). However, relative net changes were appreciable only for Artificial surfaces, which registered a substantial increase of about 50%, while Forest and semi-natural areas stayed almost constant (0.3%) and Agricultural areas slightly decreased (-4.4%). The spatial distribution of these changes was far from uniform within the territory. Urban sprawl was concentrated in the metropolitan areas of Lisbon and

Porto, as well as in central-north and south coastal areas (region of Algarve). A promoted socioeconomic development within the country, the intense rural abandonment and the development of mass tourist industry could act as main drivers for the expansion and reconfiguration of urban areas. On the other hand, in the south and interior north regions we observed a transition to vegetated land use/land cover types, probably caused by deforestation/afforestation processes and the rural abandonment. The CLC classes

mainly affected by these changes where Scrub and/or herbaceous vegetation associations, Forests and Heterogeneous agricultural areas; the increase in Artificial surfaces was precisely due to transitions from these type of land cover. The vegetated classes with higher burnt area within the RUI detected in the study period were the following: Transitional woodland-shrub, the three types of Forests considered in the CLC inventories and, the three sub-levels of Heterogeneous agricultural areas. These findings suggest the needs of extending the concept of wildland-urban interface for Portugal to rural-urban interface, defined as Forest semi-natural

plus Heterogeneous agricultural areas adjacent to Artificial surfaces.

Results of the performed analyses allow to conclude that, from 1990 to 2012, RUI increased about 70%, burnt area decreased 35% but, nonetheless, burnt area within the RUI increased 100%. These findings underline the need of frequent monitoring and assessment of land use changes and RUI evolution in Portugal, and reinforce the need to focus the attention of forest and fire managers on this highly fire prone region.

The conclusions of this study suggest and encourage more accurate analyses to characterize and map the RUI, using accurate and high-resolution data (e.g. real footprints of buildings, road network and census data). Nevertheless, our study provides precious indications as for what is the global distribution and evolution of RUI in Portugal, identifying which regions need to be prioritized in term of RUI monitoring.

**Acknowledgements**

This work was prepared in the frame of project FIREXTR - Prevent and prepare society for extreme fire events: the challenge of seeing the "forest" and not just the "trees", co-financed by the European Regional Development Fund (ERDF) through the COMPETE 2020 - Operational Program Competitiveness and Internationalization (POCI Ref: 16702) and national funds by FCT-Portuguese Foundation for Science and Technology (FCT Ref: PTDC/ATPGEO/0462/2014. The study was also supported by: i) Project Interact - Integrative Research in Environment, Agro-Chain and Technology, NORTE-01-0145-FEDER-000017, research

line BEST, co-funded by FEDER/NORTE 2020; and, ii) European Investment Funds by FEDER/COMPETE/POCI– Operacional Competitiveness and Internationalization Programme, under Project POCI-01-0145-FEDER-006958 and National Funds by FCT - Portuguese Foundation for Science and Technology, under the project UID/AGR/04033/2013. We are especially grateful to ICNF for providing the fire data and to João Pereira for the final spelling and grammar review of the manuscript.

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

**Annexes/supplementary material**

| | Road network and associated spaces | Temporary dryland crops | Temporary irrigated crops | Vineyards | Olive groves | Permanent pastures | Agroforestry systems (AFS) of cork oak with pastures | AFS o holm oak with pastures | Forests of Cork oak | Eucalypt forests | Pinus pinaster forests | Eucalyptus forests with other coniferous/softwood | Pinus pinaster forest with broadleaved/hardwood | Dense shrubs | Less dense shrubs | Dense sclerophyllous vegetation | Open forests of pinus pinaster | Natural water courses | Dam reservoirs |
|---|---|---|---|---|---|---|---|---|---|---|---|---|---|---|---|---|---|---|---|
| 211 Non-irrigated arable land | 3% | 19% | 2% | 1% | 9% | 10% | 4% | 15% | 3% | 2% | 2% | 0% | 0% | 4% | 2% | 1% | 1% | 1% | 5% |
| 212 Permanently irrigated land | 4% | 55% | 8% | 1% | 4% | 4% | 2% | 5% | 2% | 2% | 2% | 0% | 1% | 0% | 0% | 0% | 0% | 1% | 1% |
| 213 Rice fields | 6% | 16% | 16% | 0% | 1% | 7% | 7% | 8% | 9% | 1% | 1% | 0% | 0% | 0% | 0% | 0% | 0% | 3% | 0% |
| 221 Vineyards | 5% | 31% | 4% | 9% | 8% | 4% | 1% | 3% | 1% | 5% | 3% | 1% | 2% | 2% | 2% | 2% | 1% | 0% | 2% |
| 222 Fruit trees and berry plantations | 7% | 23% | 2% | 2% | 8% | 4% | 0% | 1% | 0% | 3% | 1% | 0% | 1% | 3% | 2% | 10% | 0% | 3% | 3% |
| 223 Olive groves | 1% | 48% | 1% | 3% | 9% | 6% | 2% | 6% | 1% | 2% | 1% | 0% | 1% | 1% | 0% | 2% | 1% | 1% | 6% |
| 231 Pastures | 2% | 18% | 7% | 0% | 0% | 6% | 0% | 7% | 1% | 2% | 2% | 0% | 1% | 20% | 5% | 1% | 0% | 2% | 3% |
| 241 Annual crops associated with permanent crops | 5% | 31% | 3% | 1% | 4% | 3% | 0% | 2% | 1% | 9% | 5% | 5% | 2% | 5% | 1% | 2% | 3% | 1% | 1% |
| 242 Complex cultivation patterns | 7% | 17% | 4% | 2% | 5% | 4% | 1% | 3% | 2% | 10% | 7% | 2% | 2% | 5% | 2% | 2% | 3% | 1% | 2% |
| 243 Land principally occupied by agriculture, with | 6% | 15% | 3% | 6% | 3% | 3% | 1% | 1% | 2% | 7% | 7% | 2% | 2% | 9% | 3% | 4% | 4% | 2% | 3% |
| 244 Agro-forestry areas | 1% | 37% | 1% | 0% | 6% | 13% | 5% | 9% | 5% | 1% | 0% | 0% | 0% | 0% | 0% | 0% | 0% | 1% | 7% |
| 311 Broad-leaved forest | 3% | 17% | 1% | 1% | 3% | 9% | 3% | 11% | 4% | 9% | 2% | 1% | 1% | 3% | 1% | 2% | 1% | 1% | 8% |
| 312 Coniferous forest | 7% | 2% | 5% | 4% | 1% | 1% | 1% | 0% | 2% | 6% | 16% | 2% | 3% | 14% | 2% | 1% | 11% | 1% | 2% |
| 313 Mixed forest | 8% | 4% | 7% | 4% | 1% | 1% | 2% | 0% | 3% | 20% | 8% | 5% | 3% | 7% | 2% | 1% | 2% | 0% | 2% |
| 321 Natural grasslands | 5% | 15% | 2% | 1% | 3% | 6% | 0% | 1% | 0% | 1% | 2% | 0% | 0% | 23% | 11% | 3% | 1% | 1% | 5% |
| 322 Moors and heathland | 4% | 6% | 1% | 5% | 2% | 1% | 0% | 0% | 0% | 3% | 6% | 0% | 1% | 35% | 12% | 0% | 4% | 0% | 1% |
| 323 Sclerophyllous vegetation | 3% | 12% | 1% | 0% | 3% | 8% | 0% | 4% | 3% | 3% | 0% | 0% | 0% | 0% | 0% | 17% | 0% | 10% | 8% |
| 324 Transitional woodland-shrub | 4% | 18% | 2% | 3% | 2% | 4% | 1% | 3% | 3% | 8% | 9% | 2% | 1% | 7% | 3% | 3% | 2% | 2% | 3% |
| 331 Beaches, dunes, sands | 1% | 3% | 8% | 0% | 0% | 0% | 0% | 0% | 0% | 0% | 30% | 0% | 0% | 1% | 1% | 2% | 1% | 11% | 0% |
| 332 Bare rocks | 0% | 1% | 1% | 0% | 1% | 0% | 0% | 0% | 0% | 1% | 1% | 0% | 0% | 41% | 24% | 2% | 1% | 0% | 1% |
| 333 Sparsely vegetated area | 2% | 1% | 1% | 0% | 1% | 0% | 0% | 0% | 0% | 1% | 3% | 0% | 0% | 35% | 35% | 1% | 3% | 0% | 1% |
| Sum | 4% | 22% | 3% | 2% | 4% | 6% | 2% | 5% | 3% | 6% | 5% | 1% | 1% | 6% | 2% | 2% | 2% | 1% | 4% |

**Table 3 – Fraction of COS2007v2.0 classes' area within CORINE land cover classes.**

| Area | < 1 ha | < 5 ha | 5 – 20 ha | 5 – 100 ha | >20 ha |
|---|---|---|---|---|---|
| Forest owners (%) | 31% | 30% | 14% | 10% | 15% |
| Area (%) | 10% | 16% | 12% | 7% | 55% |
| Main tree species | Maritime pine | Maritime pine and chestnut | Eucalyptus | | Holm oak and cork oak |
| Investment | No investment | | With investment | | |
| Management practices | No active management | Management depends on how economy goes | Active management | | |
| Income | Property-reserve irregular income | | Forest-enterprise | | |
| Region | Northern and central | | | | Southern |

**Table 4 - Main characteristics of a sample of forest holdings and forest owners studied by Baptista and Santos (2005). Adapted from Feliciano et al. (2015).**