# Peer review of "Global assessment of rural-urban interface in Portugal related to land cover changes"

_Natural Hazards and Earth System Sciences, 2017_

## Referee Comment (RC1) · Anonymous Referee #1 · 19 Dec 2017

This paper focuses on an interesting topic, which is undoubtedly highly relevant (both for researchers and forest managers) in all European Mediterranean countries.

The manuscript is generally well structured. It can't be accepted, however, in its current version. It requires major revisions.

First of all, the manuscript requires a substantial grammatical revision. It is often incorrectly written and there are some major issues with English (verb tenses, vocabulary, etc.). This reviewer did not have time to review the style and grammatical aspects in detail. The authors should hire an English-proofreading expert in order to substantially improve the current text. Besides, many sentences are too vague, even confusing, and should be rewritten.

[Figure]

All sections need to be substantially improved. (See detailed comments in the revised PDF file.)

The "Data and Methodology" section has to be completed. Several explanations appearing in the "Results" section have to be moved to the Methodology section. Many aspects need to be further explained and some methodological approaches have to be further justified. The characteristics and limitations of the various databases are not always well explained. In the case of the CORINE inventory, in particular, some of its limitations should have been commented (and slightly discussed in the discussion).

The "Discussion" is quite interesting, although some parts should be reduced and several relevant aspects are missing. The authors do not explain, for instance, which major habitats or plant communities correspond in Portugal to the CORINE classes that they cite throughout the manuscript. We miss this specific information (linking the broad CORINE classes to real habitats or vegetation types), which would have probably allowed to discuss other relevant issues that the paper is omitting (e.g. biodiversity, only briefly mentioned in the conclusions). The Portuguese legislation in relation to RUIs is not commented and this is a critical issue. The authors do not explain either which are the treatments usually implemented by Portuguese forest managers in RUIs and if these practices have changed in the last years due to RUI expansion and fire regime dynamics. Moreover, the discussion does not sufficiently connect the results of this research with Portuguese forest managers' needs and priorities. The authors could maybe propose some broad landscape management guidelines in relation to the objective of minimizing the risk of large intense fires under climate change.

Please also note the supplement to this comment:
https://www.nat-hazards-earth-syst-sci-discuss.net/nhess-2017-359/nhess-2017-359-RC1-supplement.pdf

[Figure]

**Supplement:**

[revised manuscript text omitted]

---

## Referee Comment (RC2) · Anonymous Referee #2 · 20 Dec 2017

Global assessment of land cover changes and rural-urban interface in Portugal by M. Tonini, J. Parente, M. Pereira

General comments

This paper intends to present a quantitative global study on evolution of both land cover/land use changes and wildland /rural urban interface in Portugal from 1990 to 2012, to analyze spatial distribution, and to characterize burned areas within the wildland /rural urban interface during the same period of time.

The study focuses on an interesting field of research, which has increasingly gained attention in recent years and it could represent a first step for more accurate analyses in order to better understand relationships among socio-economic factors, land use

changes and fire occurrence.

The topic is appropriate for readers of the Journal and the potential research outcomes may have operational value for Mediterranean Countries. The overall presentation is well structured, the methods are appropriate, discussion of results is interesting. However, the MS has some weak points and consequently it needs to be improved.

My recommendation is that the manuscript could be accepted for publication by the journal Natural Hazard and Earth System Sciences with some revision outlined below.

Specific comments

The MS would benefit from an English language revision, some sentence are not clear and should be rephrased.

The authors should select and better clarify objectives. In fact, it is not clear if the main aim of this study was to assess the impact of land use changes on burned areas, to provide a global assessment of land cover changes, or to assess the evolution of RUI.

Page 1, lines 28-30: Three important points of view are condensed in a single short sentence. The authors should either give some details on different aspects of fire problem or to delete the whole sentence, it does not seem essential for introducing paper topics.

Page 3, lines 6-9: I was not able to find coherence in this sentence, i.e. why is "spatial extension of the WUI" determined by the factors above-mentioned?

Page 3, lines 21-23: Authors should explain if WUI and RUI terms have different meanings and, in this case, why they chose RUI.

Please check citation style.

On the whole the authors should give some more details on methodology.

Table 1: Authors showed in table 1 first and second level classes of CLC but in the

[Figure]

4.2 section they discussed results relative to Corine Land Cover third level. I suggest adding CLC 3rd level in table 1.

Page 5, line 24: I suggest adding a figure where an example of RUI map at local scale is shown. This could help reader to better understand which land cover classes were included in RUI and how RUI was mapped.

Page 6, line 4: explain in the text what AA means.

Page 6, line 8-9: "predominating in the inner northern region and especially in the southern half of the country" the meaning of this sentence is not clear, please rephrase.

Page 6 line 10, and page 7 figure 5 and 6: in order to avoid confusion in reading the results reported in figures 5 and 6, I suggest adding in "Data and methodology" section a description of approach authors followed for calculating area gained or lost and net percentage changes.

Page 6, Figure 4: could authors improve resolution of this figure?

Page 9, lines 7-8: what total do authors refer?

Page 9 lines 10: authors should give some further details in the text. Reader has to look for CLC classes and codes in table 1, to calculate the sum and to compare it with the other burned areas.

Page 9 lines 11-12: I was not able to find a logical connection between this sentence and the previous one. Please explain better your thought.

Figure 8: it is not clear what percentage authors refer to.

Page 13, lines 1-3: authors should explain how the factors listed in this paragraph have affected RUI changes.

Page 13, lines 34-39: conclusions reported in this paragraph are not arising from results of this study. It would be better to move this paragraph to introduction or discus-

sion section.

Page 13, line 43: "identify which regions need to be prioritized in term of . . . . . . . . ." I do not think that this issue was addressed in this study.
* * *

---

## Author Response (AR1)

Authors thanks the reviewer for her/his time and constructive comments and suggestions, which we believe have improved the manuscript by making it more clearly and consistent. Our answers to the more general Reviewer' suggestions were uploaded in the form of a supplement.

**Comment from Referee1**

*First of all, the manuscript requires a substantial grammatical revision. The authors should hire an English-proofreading expert in order to substantially improve the current text. Besides, many sentences are too vague, even confusing, and should be rewritten.*

**Author's response**

Authors changed/rewrote all sentences highlighted by the reviewer to make them more comprehensible. An English-proofreading expert has revised the entire manuscript.

**Comment from Referee1**

*All sections need to be substantially improved. (See detailed comments in the revised PDF file.)*

**Author's response**

Authors carefully improved all sections, as can be seen in the annexed document.
In particular: (i) we propose to change the **title** as "Global assessment of rural-urban interface in Portugal related to land cover changes "; (i) in the **abstract** we introduced first the RUI and its relation with LULCC, and then the burnt area (in fact, forest first are not directly investigated in the present study); (iii) we reorganized a little bit the **Introduction** to better explain the objectives of the present paper; (iv) we moved some explanation from **Results** to **Data and Methodology** (see below); (v) we removed some long sentences from **Discussion**, expressing them in a more synthetic way, but we added information to link the broad CORINE classes to real habitats or vegetation types (see below); (vi) we reformulated the **Conclusions** (see below).

**Comment from Referee1**

*The "Data and Methodology" section has to be completed. Several explanations appearing in the "Results" section have to be moved to the Methodology section. Many aspects need to be further explained and some methodological approaches have to be further justified.*

**Author's response**

We implemented the "**Data and Methodology**" section and we moved some explanations from "**Results**" to this section.
More in detail: (i) we provided in "Data and Metgodology" section a new and more complete version of Table 1, showing the CORINE Land Cover nomenclature for the three level; (ii) the concept of "Area gained and lost" and "Net Chenges" was detailed and the computation of these values was well described; (iii) The choice of the buffer width used to compute the RUI has been discussed and justified; (iv) the CLC hierarchical levels considered for each analyses was deeper explained and justified based on the objectives.

**Comment from Referee1**

*The characteristics and limitations of the various databases are not always well explained. In the case of the CORINE inventory, in particular, some of its limitations should have been commented (and slightly discussed in the discussion).*

**Author's response**

Authors addressed to this issue in the section "**Discussion**"

**Comment from Referee1**

*The "Discussion" is quite interesting, although some parts should be reduced and several relevant aspects are missing. The authors do not explain, for instance, which major habitats or plant communities correspond in Portugal to the CORINE classes that they cite throughout the manuscript. We miss this specific information (linking the broad CORINE classes to real habitats or vegetation types), which would have probably allowed to discuss other relevant issues that the paper is omitting (e.g. biodiversity, only briefly mentioned in the conclusions).*

**Author's response**

We reduced this session, namely we removed details from literature on urban growth of MAP and MAL. As regards the request of including specific information linking CLC classes to real habitat, author decided to include in the analyses a more accurate and detailed Portuguese land use map, namely the Soil Use and Occupancy Chart (*Carta de Uso e Ocupação do Solo*, COS). The description of this map was added in the section "Data and Methodology". We compared CLC2006 and COS2007v2.0 because these are the closest inventories (in time) between them and within the study period. We introduced a table showing the result of the overlapping between the two land use maps, allowing to identify the vegetation types/major habitats/plant communities in each of the CLC for Portugal. This paragraph was added in the section "Discussion".

**Comment from Referee1**

*The Portuguese legislation in relation to RUIs is not commented and this is a critical issue. The authors do not explain either which are the treatments usually implemented by Portuguese forest managers in RUIs and if these practices have changed in the last years due to RUI expansion and fire regime dynamics. Moreover, the discussion does not sufficiently connect the results of this research with Portuguese forest managers' needs and priorities. The authors could maybe propose some broad landscape management guidelines in relation to the objective of minimizing the risk of large intense fires under climate change.*

**Author's response**

As far as we know, there is no specific or general Portuguese legislation about WUI or RUI. In Portugal, there is only one general mention about WUI/RUI in the National Plan to Protect the Forests against Wildfires (CM, 2009). In this Plan it is suggest that to protect urban-forest

interface it will be necessary to create and maintain an external buffer strips around population clusters, especially in those with the highest fire vulnerability, as well as around parks, industrial polygons, landfills, housing, shipyards, warehouses, and other buildings. Usually, it is suggest a buffer of 100 meters around population clusters, 10 meters for each side of a road, and 50 meters around houses. For private and communal property there are the Municipal Plan for Territorial Planning, which include the municipal master plan, that regulates all land uses, the urbanization plan, and others specifics (Feliciano et al., 2015). Private properties within protect areas are restrict by SPFP. At local level, there is the Municipal director plan for landscape plan, and which incorporates the new municipal plan for forests against fire since 2006.

Authors introduced all these aspects in the new version on the present manuscript, in the section "**Discussions**".

Finally, better quality figures have been produced and will be uploaded separately.

Authors thanks the reviewer for her/his time and constructive comments and suggestions, which we believe have improved the manuscript by making it more clearly and consistent. Our answers to the more general Reviewer' suggestions were uploaded in the form of a supplement.

**Comment from Referee2**
*The MS would benefit from an English language revision, some sentence are not clear and should be rephrased.*

Authors changed/rewrote all the sentences highlighted by the reviewers to make them more comprehensible. An English-proofreading expert has revised the entire manuscript.

**Comment from Referee2**
*The authors should select and better clarify objectives. In fact, it is not clear if the main aim of this study was to assess the impact of land use changes on burned areas, to provide a global assessment of land cover changes, or to assess the evolution of RUI.*

We reorganized the **Abstract** and the **Introduction** to better explain the objectives of the present paper, and we specified at the end of Introduction: "In the present study, authors investigate the RUI in Portugal: the main objective is to analyze changes in land use/land cover occurred in this country in the period 1990-2012 and to assess their impact on RUI's evolution. Moreover, a qualitative and quantitative characterization of burnt areas within the RUI in relation to the LULCC is provided. Finally, this research provides a first attempt to map the RUI's extension at national level for continental Portugal."

**Comment from Referee2**
*Page 1, lines 28-30: Three important points of view are condensed in a single short sentence. The authors should either give some details on different aspects of fire problem or to delete the whole sentence, it does not seem essential for introducing paper topics.*

We moved this sentence later where it make more sense.

**Comment from Referee2**
*Page 3, lines 6-9: I was not able to find coherence in this sentence, i.e. why is "spatial extension of the WUI" determined by the factors above-mentioned?*
*We reformulated this sentence*

We changes in "these factors are broadly considered to elaborate WUI maps."
This is also clear from the cited literature.

**Comment from Referee2**
*Page 3, lines 21-23: Authors should explain if WUI and RUI terms have different meanings and, in this case, why they chose RUI.*

We added: *"*In this respect, recent studies defined the Rural-Urban Interface (RUI) as an alternative to the WUI, to highlight the importance of including the rural area,… In the present study, authors investigate the RUI in Portugal"

**Comment from Referee2**
*Please check citation style.* Done

**Comment from Referee2**
*On the whole the authors should give some more details on methodology.*

We implemented the "**Data and Methodology**" section and we moved some explanations from "**Results**" to this section.
More in detail: (i) we provided in "**Data and Metgodology**" section a new and more complete version of Table 1, showing the CORINE Land Cover nomenclature for the three level; (ii) the concept of "Area gained and lost" and "Net Chenges" was detailed and the computation of these values was well described; (iii) The choice of the buffer width used to computed the RUI has been discussed and justified; (iv) the CLC hierarchical level considered for each analyses was deeper explained and justified based on the objectives.

*Table 1: Authors showed in table 1 first and second level classes of CLC but in the 4.2 section they discussed results relative to Corine Land Cover third level. I suggest adding CLC 3rd level in table 1.* Done

*Page 5, line 24: I suggest adding a figure where an example of RUI map at local scale is shown. This could help reader to better understand which land cover classes were included in RUI and how RUI was mapped.*

We better explain the procedure to map the RUI in the section "**Data and Methodology**": *"*RUI was then mapped for each period using a geospatial approach designed to extract the area of intersection between a buffer around the Artificial Surfaces (AS) and the area resulting from the sum of the Forest and Semi-Natural Area (FSNA) plus the Heterogeneous Agricultural Areas (HAA). Different buffer width from 100 m to 2000 m were tested, but finally we adopted a buffer width of 1 km, corresponding to two times the spatial resolution of the CORINE Land Cover inventory (that is 500 by 500 m): this value is in line with values applied in other countries for WUI mapping (Vilar 2016, Radeloff et al., 2005) and, in the same time, is enough large to avoid bias in the results. The others agricultural areas (i.e. arable lands, permanents crops and pastures) were not included in the RUI definition since these vegetated land covers are usually well managed, mostly irrigated and frequently constitute an

obstruction to fire spread. Similarly, San-Miguel-Ayanz et al., (2012) suggested that HAA have to be considered in the definition and quantification of the rural-urban interface in Portugal, together with forest and semi-natural areas. The geocomputation which allowed producing the RUI's maps was performed under ArcGIS™ software environment. Namely, the geoprocessing workflows was implemented into a Model Builder (Fig.3), a specific application used to create, edit, and manage models, meant as workflows that string together sequences of geoprocessing tools (e.g. selection, buffer, intersect), feeding the output of one tool into another tool as input (i.e. the raster or vector spatial data). "

*Page 6, line 4: explain in the text what AA means.* Done

*Page 6, line 8-9: "predominating in the inner northern region and especially in the southern half of the country" the meaning of this sentence is not clear, please rephrase.* Done

*Page 6 line 10, and page 7 figure 5 and 6: in order to avoid confusion in reading the results reported in figures 5 and 6, I suggest adding in "Data and methodology" section a description of approach authors followed for calculating area gained or lost and net percentage changes.* Done

*Page 6, Figure 4: could authors improve resolution of this figure?* Done: all the figures were uploaded in high resolution as single files

*Page 9, lines 7-8: what total do authors refer?*
We specified that it refers to the total burnt area

*Page 9 lines 10: authors should give some further details in the text. Reader has to look for CLC classes and codes in table 1, to calculate the sum and to compare it with the other burned areas.*
We changed as : ". It also emerges that the three over four sub-level of heterogeneous agricultural areas (code 243, 241, 242) are highly affected by wildfires"

*Page 9 lines 11-12: I was not able to find a logical connection between this sentence and the previous one. Please explain better your thought.*
We changed as "thus confirming the need of including HAA in the definition of the RUI".
Moreover, since in methodology we better explained the CLC classes we used to map the RUI, we hope that this sentence is now clearer.

*Figure 8: it is not clear what percentage authors refer to.*
We refer to the percentage over the total BA for each frame period. We better explained this both in the image caption and in the text.

*Page 13, lines 1-3: authors should explain how the factors listed in this paragraph have affected RUI changes.*

We removed this sentence and we and the following : "These changes could be associated with the relative decrease of BA in the last investigated period, as a consequence of recent plans for territorial spatial planning and protection of forest against forest fires (Mateus and Fernandes, 2014; Parente et al., 2016)."

*Page 13, lines 34-39: conclusions reported in this paragraph are not arising from results of this study. It would be better to move this paragraph to introduction or discussion section.*
We deleted the second part of this paragraph (line 37-39) and we moved the first sentence on to "**Introduction**".

*Page 13, line 43: "identify which regions need to be prioritized in term of : : :: : :: : :" I do not think that this issue was addressed in this study.*
We changed as : "and identify which areas need to be prioritized in term of RUI monitoring". These are in fact the mapped RUI area.

Finally, better quality figures have been produced and will be uploaded separately

[revised manuscript text omitted]

. The main changes in terms of surface arewere registered by AS, which increaseincreased 165×10$^3$ ha, and AA, which

decreasedecreased 184×10$^3$ ha. These values have a completely different impact, but in terms of net percentage of change: the net

increase of AS is ofwas about 50%, while the net decrease of AA isdecreased only of 4.4%. The two classes which manly contributed

to the increase in AS arewere AA, with 110×10$^3$ ha, and FSNA, with 50×10$^3$ ha.

[Figure]

[Figure]

Field Code Changed

**Figure 4 – Map of land cover/land use transition from 1990 and 2012, evaluated considering the first level hierarchy of CLC 1990 and CLC 2012**

[Figure]

**Figure 5 – Area lost and gained from 1990 to 2012 for each CORINE land cover classes, considering the first level hierarchy. Net percentage changes were computed relatively to the total area of each class in the later land cover.**

A more detailed analyses was carried out to investigate changes from 1990 to 2012 occurred within classes considering the  second level hierarchy . Figure 6 shows that the majority of the CLC classes (level 2, Table 1) displayed important net changes in terms of relative gains and losses compared with values for the same classes in the later period. Scrub and/or herbaceous vegetation associations (code 32) registered a net gain of about 520 $\times 10^3$ ha (+24%), while the Forest area (code 31) decreased

about 460 ×10³ ha (-23%). Arable land (code 21) was the only  AA registering an important negative net change of -225 ×10³ ha (-20%). Among AS, Urban fabric (code 11) significantly increased 110 ×10³ ha (45%), and, in terms of net percentage of change by class, all the other three AS sub-levels, including Industrial/commercial and transport unit (code 12), Mine/dump and construction sites (code 13), Artificial/non-agricultural vegetated areas (code 14), increased more than half.

10

| CLC level 1 | | CLC level 2 | |
|---|---|---|---|
| 1 | Artificial surfaces | 1.1 | Urban fabric |
| | | 1.2 | Industrial, commercial and transport units |
| | | 1.3 | Mine, dump and construction sites |
| | | 1.4 | Artificial, non-agricultural vegetated areas |
| 2 | Agricultural areas | 2.1 | Arable land |
| | | 2.2 | Permanent crops |
| | | 2.3 | Pastures |
| | | 2.4 | Heterogeneous agricultural areas |
| 3 | Forest and semi natural areas | 3.1 | Forests |
| | | 3.2 | Scrub and/or herbaceous vegetation associations |
| | | 3.3 | Open spaces with little or no vegetation |
| 4 | Wetlands | 4.1 | Inland wetlands |
| | | 4.2 | Maritime wetlands |
| 5 | Water bodies | 5.1 | Inland waters |
| | | 5.2 | Marine waters |

Table 1 - CORINE land cover (CLC) first and second level hierarchy

[Figure]

**Figure 6 – Area lost and gained and lost from 1990 to 2012 for each CORINE land cover classes, considering the second level hierarchy. Net percentage changes arewere computed relatively to the total area of each class in the later land cover.**

The bar graph with the contributions to net changes in the AS sub-levels (Fig. 7) shows that Urban fabric, (orange bars), which  includes buildings, roads and artificially surfaced areas, grew at the expense almost exclusive of HAA (code 24). On the other hand, the increase of Industrial commercial and transport  (blues bars) was mainly due to the decrease of Forests (code31), HAA (code 24) and Scrub and/or herbaceous vegetation associations. (code 32).

[Figure]

[Figure]

**Figure 7 – Contribution to the net changes from 1990 to 2012 of " "Industrial, commercial and transport"  from the**

       other CLC sub-levels

**4.2 Spatial distribution and characterization of burned areas**

Almost all the CLC third-level classes belonging  FSNA (code 3) were affected by
     wildfires in terms of

and semi-naturalburned area (Table 12), with the Transitional woodland-shrub (code 324) and Mixed forest (code 313) as the first and second more damaged classes in each period.. This trend iswas similar in all the four investigated frame-periods, as highlighted in Fig. 8, which reveals that the sum of burnt 8, where the same results are expressed in percentage for each CLC classes considering only the areas within these two classes exceed the 50%RUI, as the ratio of BA over the total. BA for the entire frame period. The peak of BA in Transitional woodland-shrub equal(code 324) equals to about 43 ×10³ ha in the period 2005-2007, compared with about 15×10³ ha in 2011-2013, 14×10³ ha in 1999-2001 and

[revised manuscript text omitted]